# Evaluation of Lab-on-a-Disc Technique Performance for Soil-Transmitted Helminth Diagnosis in Animals in Tanzania

**DOI:** 10.3390/vetsci11040174

**Published:** 2024-04-13

**Authors:** Sarah L. Rubagumya, Jahashi Nzalawahe, Gerald Misinzo, Humphrey D. Mazigo, Matthieu Briet, Vyacheslav R. Misko, Wim De Malsche, Filip Legein, Nyanda C. Justine, Namanya Basinda, Eliakunda Mafie

**Affiliations:** 1Department of Microbiology, Immunology and Parasitology, College of Health and Allied Sciences, St. Joseph University in Tanzania, Dar es Salaam P.O. Box 11007, Tanzania; 2Department of Veterinary Microbiology, Parasitology and Biotechnology, College of Veterinary Medicine and Biomedical Sciences, Sokoine University of Agriculture, Morogoro P.O. Box 3019, Tanzania; nzalawahej@sua.ac.tz (J.N.); gerald.misinzo@sacids.org (G.M.); eliakunda.mafie@sua.ac.tz (E.M.); 3SACIDS Foundation for One Health, Sokoine University of Agriculture, Morogoro P.O. Box 3297, Tanzania; 4Department of Medical Parasitology, School of Medicine, Catholic University of Health and Allied Sciences, Mwanza P.O. Box 1464, Tanzania; humphreymazigo@bugando.ac.tz (H.D.M.); justinenyanda@bugando.ac.tz (N.C.J.); n.basinda@bugando.ac.tz (N.B.); 5µFlow Group, Department of Chemical Engineering, Vrije Universiteit Brussel, 1050 Brussels, Belgium; matthieu.briet@vub.be (M.B.); veaceslav.misco@vub.be (V.R.M.); filip.legein@vub.be (F.L.)

**Keywords:** soil-transmitted helminths, domestic pigs, dogs, flotation, McMaster, LoD

## Abstract

**Simple Summary:**

Soil-transmitted helminth parasites are associated with a neglected tropical disease affecting both humans and animals. Proper management of the infection requires an accurate diagnostic technique. Therefore, in an era of STH prevention and control, a new lab-on-a-disc (LoD)-based diagnostic technique called single-image parasite quantification (SIMPAQ) was developed with the ability to focus all STH eggs in an imaging zone, where a digital image can be captured for further processing. This report evaluates the performance of SIMPAQ in terms of prevalence, sensitivity, specificity and predictive values. In this study, a total of 518 animal faeces samples were examined using McMaster, test tube simple flotation, and the LoD technique. The highest prevalence was demonstrated by the LoD technique, which also demonstrated high sensitivity in the detection of STH eggs.

**Abstract:**

Soil-transmitted helminth (STH) infections are caused by roundworms, hookworms, whipworms, and thread worms. Accurate diagnosis is essential for effective treatment, prevention, and control of these infections. This study evaluates a new diagnostic method called Single-image Parasite Quantification (SIMPAQ), which uses a lab-on-a-disc (LoD) technique to isolate STH eggs into a single imaging zone for digital analysis. The study evaluates the purification performance of the SIMPAQ technique for detecting STH eggs in animal samples. This was a cross-sectional study conducted among 237 pigs and 281 dogs in the Morogoro region in Tanzania. Faecal samples were collected and processed with the LoD technique, as well as flotation and McMaster (McM) methods for comparison purposes. The overall prevalence of STH infections was high as per the LoD technique (74%), followed by McM (65.44%) and flotation (65.04%). Moreover, the overall performance of the LoD technique, using McM as the gold standard, was 93.51% (sensitivity), 60.89% (specificity), 81.91% (PPV), and 83.21% (NPV). The LoD technique exhibited high prevalence, sensitivity, and NPV, which demonstrates its value for STH egg detection and its crucial role in the era of accurate STH diagnosis, promoting proper management of the infection.

## 1. Introduction

Soil-transmitted helminth (STH) infections in domestic pigs and dogs are caused by a group of nematode parasitic worms which include the following: *Ancylostoma*, *Trichuris*, *Toxocara*, Ascarid, and *Oesophagostomum* species. STH infections in these animals affect their productivity and performance by damaging of some organs and absorbing essential nutrients, thus leading to anaemia, poor growth, and occasionally causing death [1,2]. Dogs and pigs are also known to harbour STHs reported to be zoonotic in different parts of the world. These zoonotic STHs include hookworms: *Ancylostoma* and *Toxocara* species in dogs, as well as *Ascaris* and *Trichuris* species in pigs [3,4,5,6]. The accurate diagnosis of STH infection plays a very important role in the treatment, prevention, and control of the parasites. Tentative diagnosis of STH could be achieved based on the clinical signs presented by the animals. However, the use of these signs alone for routine diagnosis is unreliable, as most helminth infections in animals present similar symptoms. Therefore, laboratory confirmation is mandatory for identifying adult worms, worm fragments, eggs, and larvae in order to achieve accurate diagnosis [2,7].

Conventional parasitological laboratory methods widely used for the routine diagnosis of STH in animals are either qualitative or quantitative. Qualitative methods include direct faecal smear and faecal flotation, which are used to check the presence or absence of parasites in faeces. Direct faecal smear involves spreading a wet mount preparation directly onto the microscopic slide using a small amount of faeces. It is used to detect the presence of motile parasites and eggs of helminths in faeces. This method is very simple, timesaving, requires minimal equipment, and does not destroy the parasite eggs nor the larvae. However, this method does result in a significant amount of debris on the microscopic slide and has very low sensitivity; thus, it is not recommended for routine faecal examination [4]. Faecal flotation may either be simple flotation or test tube flotation [2]. The basic principle of faecal flotation consists of using flotation fluid which has a higher density than that of eggs. As a result, these eggs tend to float on the flotation medium. The advantage of faecal flotation is simple and inexpensive, since this method concentrates the parasites on the surface of the flotation media; it is more sensitive compared with direct faecal smear. However, this technique is less sensitive to low and moderate STH infection and could not be used for estimating worm burden in animals [8].

The McMaster technique is a quantitative method that was designed to estimate the number of eggs per gram of faeces. The McMaster method relies on the McMaster slide, made up of two glass or plastic slides with two gridded chambers, which are filled with a solution composed of faeces and the flotation fluid. The worm eggs float towards the surface of the McMaster chambers, where they can be observed and counted with the aid of a microscope [9]. To calculate the eggs per gram (EPG), the number of eggs observed in the two chambers is multiplied by 50. Alternatively, the EPG can also be calculated by multiplying the number of eggs observed in one chamber by 100. McMaster has simple procedures and provides rapid results. However, the McMaster method is relatively inaccurate as the sensitivity is limited to 50 EPG and can only identify worm genera with morphologically distinct ova [2]. Efforts to develop techniques which address the shortfalls of the McMaster methods are ongoing. For instance, the FLOTAC technique has been developed, which relies on centrifugal flotation and has the capacity to detect one egg per gram of faeces. FLOTAC apparatus, a centrifuge, and a microscope are the main pieces of equipment required. FLOTAC apparatus is cylindrical and has two floating chambers of 5 ml in volume capable of holding up to 1 g of faecal sample for microscopic analysis. This method is highly sensitive and accurate and can be performed for fresh and preserved faecal sample in human and animals. However, the main limitation of FLOTAC technique is the need for a large-volume centrifuge with a rotor for FLOTAC apparatus, which is not always available in most laboratory settings in developing countries [10].

To complement efforts for the development of reliable diagnostic methods, single-image parasite quantification (SIMPAQ) lab-on-disc (LoD) technology has been developed [11,12,13,14]. SIMPAQ is based on the centrifugation and flotation of parasite eggs, which focuses them in one imaging zone; a single digital image can be captured for further processes, such as the quantification and identification of eggs. This method has four new features that are important for facilitating an accurate diagnosis. The first feature is guided two-dimensional (2D) flotation: centrifugation and natural flotation. To accelerate the motion of the eggs toward the centre where the imaging zone is present, centrifugation forces are applied. Simultaneously, with the aid of the natural buoyancy force caused by the Earth’s gravity, eggs move on top of the chamber in a perpendicular direction while moving toward the centre. The converging chamber is the second feature of SIMPAQ. The flow chamber decreases in height and width as it approaches the centre to facilitate the monolayer of eggs packed in the imaged zone for counting and identification. Another feature is continuous size-based filtering: floating material larger than eggs is trapped prior to reaching the imaging zone, while that smaller than eggs is trapped beyond the imaging zone. The last feature is single-shot imaging: parasite eggs are collected in the collection chamber (the image zone) as a packed monolayer, allowing the user to capture a single shot which can be used for egg counting (manual or digital) and egg identification [14].

In the era of STH prevention and control, accurate diagnosis is a crucial factor to consider. However, a variety of challenges arise with conventional diagnostic methods, including decreased sensitivity to low infection intensities. Therefore, a new LoD technique, SIMPAQ, was developed facilitate diagnosis in the field and laboratory settings. This study was designed to evaluate the performance of the centrifugation-mediated flotation technique on a large number of samples collected from naturally infected animals.

## 2. Materials and Methods

### 2.1. Study Area

This study was conducted in two districts of the Morogoro region: Mvomero and Morogoro municipalities. The estimated population in the Morogoro region was 3,197,104 according to the Tanzania population and household census conducted in 2022. The region experiences moderate temperature and rainfall; the annual average temperature ranges from 18 °C to 30 °C, and average rainfall ranges between 600 mm and 1800 mm. The rainfall pattern is bimodal. Heavy precipitation is experienced from March and May, with light rains between November and January. The major economic activities in Morogoro region are agriculture, farming, and keeping livestock. The selection of districts, divisions, wards, village/streets, and animals was based on simple random sampling. Mvomero District lies at latitude 06°26′ south and longitude 37°32′ east. It is bordered by Tanga region (north), Pwani region (northeast), Morogoro Rural District (east), Morogoro urban (southeast), and Kilosa District (west). Based on Tanzania population and household census, Mvomero District is divided into 4 divisions with 30 wards and 115 villages. Two divisions (Mzumbe and Mgeta) and 4 wards (Nyandila, Mgeta, Lingali, and Mlali) were visited in Mvomero District. Morogoro municipality is found at latitude 06°49′ south and longitude 37°39′ east. It is located on the lower Uluguru Mountain slopes with a peak of about 500 to 600 m above sea level. It has 29 wards, and 272 streets. The wards visited in Morogoro municipality were Mazimbu, Kihonda Maghorofani, Kilakala, Bigwa, and Kichangani (Figure 1).

### 2.2. Study Design and Selection of Study Sites

This was a cross-sectional study conducted targeting randomly sampled domestic pigs and dogs in Morogoro municipal and Mvomero District. Districts and wards were randomly selected for involvement in this study. A list of villages/streets with domestic pigs and dogs was obtained from the ward office; their selection was based on simple random sampling. Similarly, at village/street level, the selection of households and animals was based on the random sampling method.

### 2.3. Sample Size Estimation

The sample size (n) was determined by using the following formula [15]:(n) = ((Z_1−α/2_)^2^ (p) (q))/d^2^

Here, ‘n’ refers to sample size, ‘Z_1−α/2_’ represents critical value, and ‘α’ is a standard value for the corresponding level of confidence (at the 95% confidence interval (CI) or 5% level of significance, α is 1.96), ‘p’ corresponds to the prevalence (presumed to be 0.5), ‘q’ corresponds to 1-p, and ‘d’ refers to the margin of error (0.05). Estimations of sample sizes for domestic pigs and dogs were achieved by using the expected prevalences reported by [16,17]. Therefore, the calculated sample sizes were 280 dogs and 222 domestic pigs. A total of 518 animals were involved in this study; 259 were from Morogoro municipality and 259 were from Mvomero. The numbers of pigs and dogs involved in this study were 237 and 281, respectively. 

### 2.4. Data Collection 

A data capture form was used to obtain information about the participating households and their animals. The information included the region, district, ward, the owner’s name, and contact details. Information related to the animal’s age, sex, deworming status, housing condition or management system (indoor/semi-intensive/free roaming), and the housing cleaning were collected. 

Prior to collection of the faecal samples, the participating households were briefed on the purpose of the study. The randomly selected animals at household level were observed for some time until they defecated; then, the fresh faecal samples were collected from the ground using gloved hands. Each sample was labelled with a unique identification number which contained information on the sex, age, breed, locality, and date [14]. Samples were transported using a cool box containing ice packs to the Parasitology laboratory at the Department of Microbiology, Parasitology and Biotechnology of Sokoine University of Agriculture, where they were refrigerated (at approximately 4 °C) before being analysed.

### 2.5. Faecal Sample Processing and Analysis

#### 2.5.1. Qualitative Faecal Sample Analysis 

The samples were processed using the simple flotation method and the SIMPAQ technique. Briefly, the simple flotation method was performed by taking 3 g of faeces mixed into 50 mL of saturated salt solution of sodium chloride (NaCl) flotation fluid, whose specific gravity was 1.2. The mixture was filtered using a tea strainer into a beaker. The test tube was placed into the beaker with the mixture for 15 min before it was slowly removed and put onto the microscopic slide. A drop of the mixture from the bottom of the test tube was left on the microscopic slide, and a cover slip was put on top of the slide for examination under the microscope at 10x and 40x magnification to determine the presence or absence of STH eggs [18,19].

The LoD setup was assembled, including a microscope, a centrifuge, and a disc, as shown in Figure 2. To perform the LoD technique, 1 g of faecal sample was diluted in 20 mL distilled water in a 50 mL Falcon tube and the solution was shaken for about 20 s until the mixture became homogeneous. The mixture was then filtered through 2 filters with 200 µm and 20 µm pore sizes, respectively. Next, the 20 µm filter surface was rinsed with 2 mL distilled water to recover parasite eggs, and the rinsed solution was transferred to a 2 mL centrifugation vial. Centrifugation of the mixture was carried out for 3 min at 1500 revolutions per minute (rpm), and the sediment was re-suspended in 500 µL flotation solution. The mixture was transferred to a 3 mL syringe and injected into the LoD. The disc was filled with the flotation fluid prior to sample injection. The disc was allowed to settle for 5 min then centrifuged at 6000 rpm for 5 min. The disc was examined under the SIMPAQ microscope. The entire disc was observed for the presence or absence of STH eggs, and images from different chambers of the disc were taken [10]. 

#### 2.5.2. Quantitative Faecal Sample Analysis

Faecal egg count was performed using a McMaster technique, briefly, 3 g of faecal sample was mixed with 42 mL flotation fluid thoroughly and filtered using a tea strainer. A sub-sample was taken with a Pasteur pipette and filled in the McMaster counting chambers. It was then left to settle on the table for about 3–5 min before counting, allowing the eggs to float in the counting chamber. The McMaster slide was examined under a microscope, and the counting was carried out. The number of eggs obtained was multiplied by 100 to obtain the eggs per gram (EPG) of faeces [18,20]. 

### 2.6. Data Analysis

Data were entered, cleaned, and coded into Microsoft Excel Windows 10, and then uploaded into STATA statistical software version 14 for analysis. To assess the performance of the LoD technique, the McMaster and test tube simple flotation method was used as a gold standard to calculate the prevalence (P), sensitivity (Sn), specificity (Sp), negative predictive value (NPV), and positive predictive value (PPV). Additionally, the Kappa value was calculated so as to determine the strength of agreement of the diagnostic method. A *p*-value < 0.05 was considered as statistically significant [21,22].

## 3. Results

### 3.1. Prevalence of Soil-Transmitted Helminths

The obtained overall prevalence of STH varied amongst the parasitological methods. The highest overall prevalence of STH in animals (*n* = 518) was obtained by the LoD technique (74%), followed by McMaster (65.44%) and flotation (65.04%). Moreover, the highest prevalence for individual STH species was obtained using the LoD diagnostic technique, in which the most prevalent species eggs were strongyle type (72.01%), followed by ascarid (13.51%), whipworm (9.65%), and threadworm (0%). The prevalences of strongyle, ascarid, whipworm, and threadworm eggs using simple flotation method were 61.97%, 8.88%, 6.76% and 0.58%, respectively. However, the prevalence was slightly increased using the McMaster method, whereby strongyle, ascarid, whipworm, and threadworm eggs had prevalences of 62.16%, 9.46%, 7.72% and 0.58%, respectively.

However, the percentage of pigs which had at least one STH species from any of the three diagnostic techniques was 95.36%, with higher percentages by LoD (86.50%), followed by simple flotation (85.23%) and McMaster techniques (83.54%). Moreover, in dogs, the percentage was 65.84% by any of the techniques, while it was 64.77%, 50.53%, and 48.75% by LoD, McMaster and simple flotation techniques, respectively. 

The arithmetic mean faecal egg counts in dogs were 600 (95% CI: 440.79–759.21) EPG, 283.63 (95% CI: 120.00–447.26) EPG, 4.29 (95% CI: 1.18–8.79) EPG, and 2.49 (95% CI: −0.39–5.37) EPG for strongyle, ascarid, whipworm, and threadworm, respectively. Based on pigs, the arithmetic means were 876.37 (95% CI: 730.62–1022.11) EPG, 111.39 (95% CI: 52.45–170.33) EPG, and 67.93 (95% CI: −2.67–138.54) EPG for strongyle, whipworm, and ascarid eggs, respectively. The images of strongyle and ascarid eggs are shown in Figure 3.

### 3.2. Overall Diagnostic Performance of LoD Technique 

To assess the performance of a diagnostic tool, the prevalence, sensitivity, specificity, negative predictive value (NPV), and positive predictive value (PPV) need to be calculated. Sensitivity and specificity are basic characteristics of the tool, while NPV and PPV define the relevance of the test results of the corresponding specific diagnostic tool. The overall LoD technique performance for both animal species (*n* = 518) is shown in Table 1 below.

### 3.3. Diagnostic Performance of LoD Technique for STH in Domestic Pigs and Dogs

The performance was reduced for domestic pigs, whose sensitivity was 90.40%, specificity was 33.33%, PPV was 87.32%, and NPV was 40.63% (Table 2). However, the highest performance of the LoD technique was observed for dogs, with a sensitivity, specificity, NPV, and PPV of 97.87% and 68.57%, 96.97%, and 75.82%, respectively (Table 3). However, the performance was reduced for domestic pigs, whose sensitivity was 90.40% and, specificity was 33.33%, PPV was 87.32%, and NPV was 40.63% (Table 2). Furthermore, the performances of the flotation and McMaster technique for both animal species were the best, with observed sensitivity, specificity, NPV, and PPV values above 80%.

There was a slight test agreement between the LoD technique and other diagnostic techniques for STH detection in domestic pigs (ĸ = 0.2258 and ĸ = 0.2528), as shown in Table 2. The LoD technique agreed moderately (ĸ = 0.6651 and 0.6271) with other diagnostic techniques for the detection of STH in dogs (Table 3). However, there was a strong agreement between flotation and McMaster techniques (ĸ = 0.8720 and ĸ = 0.9288)

### 3.4. LoD Technique Performance for Individual STH Species in Domestic Pigs and Dogs

This study revealed that the LoD technique performance varied depending on the species of the animal and parasite type. The highest performance was observed for ascarid eggs in dogs (sensitivity = 97.14%; specificity = 93.9%), whereby in domestic pigs the performance was slightly different (sensitivity = 72.73%; specificity = 93.63%). The sensitivity for all STH species was above 70%, with the exclusion of whipworms in dogs, whose sensitivity was below 50%. However, the highest specificity was obtained for whipworm eggs (in dogs (96.4%) followed by domestic pigs (95.5%)), while the lowest was found in strongyle eggs (69.6% for dogs and 33.33% for domestic pigs) (Table 4). 

### 3.5. Overall Performance of the LoD Technique for Individual STH Species 

Strongyle were the most prevalent species isolated; their sensitivity was also the highest among all STH species, followed by ascarid and whipworm. However, the highest specificity was observed for whipworm, followed by ascarid and strongyle eggs (Table 5).

## 4. Discussion

STH infection is a neglected disease in developing countries, and can affect both animals and human beings. In animals, the infection is of veterinary importance, particularly in domestic pigs and dogs, where there is a great risk for zoonosis. Accurate and reliable diagnostic techniques are essential elements for the evaluation of treatment success and control programs for STH infections. The most commonly used diagnostic techniques in animals for STH diagnosis are flotation and McMaster techniques. However, these techniques exhibit several challenges, which has prompted the development of new diagnostic tools [2,8]. Therefore, the LoD technique was developed to address the shortfalls of these commonly used diagnostic methods. This study aimed to assess the performance of the LoD technique in comparison with the flotation and McMaster methods. Generally, the LoD technique attained high sensitivity when compared with simple flotation and McMaster techniques for the detection of STH infection in pigs and dogs.

Furthermore, the present study revealed a higher prevalence of STH infection in dogs and pigs, especially when using the LoD technique in comparison with the other parasitological methods used. The prevalence was consistently high when using the LoD technique for both animal and STH species. Considering that all animal species recruited were asymptomatic, this may have been an indication that these infections are still endemic in the study area. The arithmetic mean faecal egg count was high for strongyle type of eggs, followed by ascarid and whipworm in both pigs and dogs.

The overall sensitivity of LoD technique was high (≥93%) by using McMaster and flotation methods as reference diagnostic techniques. The LoD technique was able to capture many of STH infection cases while missing about 7%. However, the specificity was low (≈60). The low specificity of LoD may be due to the low sensitivity of our reference diagnostic techniques especially in samples with low eggs counts [23,24,25]. Since less sensitive methods were used as gold standards, the LoD technique could detect eggs in samples which were negative by the reference techniques. Those samples which were positive by the LoD method but negative by reference diagnostic methods were regarded as false positive, hence lowering the specificity of LoD technique. 

Based on the two animal species, LoD performed better for dogs in terms of sensitivity (97.87%), while the specificities were not considerably high for both animal species. The best LoD sensitivity was observed for strongyle and ascarid eggs in dogs, whose sensitivity was almost similar (97%), while the lowest sensitivity was observed for whipworm in dogs. Nevertheless, the performance of the LoD technique for the detection of STHs in domestic pigs was slightly lower compared with that of dogs. The lowest specificity was observed in domestic pigs (33.33%). The low specificity of the LoD technique may be attributed to the low ability of flotation and McMaster methods to detect eggs in low infection intensities [25]. The differences in the LoD technique performance for animal and STH species may be due to different infection intensities observed, which were measured in terms of faecal egg counts. The best LoD performance was observed for three STHs that had considerably high mean EPG. The influence of infection intensity on the sensitivity of parasitological techniques was also discussed by previous studies [26,27].

Moreover, the predictive values (positive and negative) indicate the relevance of the positive and negative results. The overall PPV and NPV for the LoD technique were above 80%; this indicates that only about 80% of the results were accurate. Comparable results were obtained for domestic pigs and dogs, although the negative predictive values in domestic pigs were very low, indicating that only about 40% of the negative results were actually accurate. The test agreement was expressed as a kappa value, which ranged between −1 and 1. For a perfect agreement, a value of 1 is obtained, while 0 is the expected agreement by chance, and negative values indicate disagreement between the tests [28]. According to the kappa scale developed by [29], the LoD technique was in moderate agreement with flotation (0.56) and McMaster (0.58) methods.

## 5. Conclusions

The newly developed lab-on-a-disc technique, single-image parasite quantification (SIMPAQ), demonstrated high sensitivity for soil-transmitted helminth detection when compared with conventional simple flotation and McMaster methods. However, the specificity was not high for both animal species. Based on these findings, the use of the LoD technique along with the present conventional technique for the detection of STH eggs in animal faecal samples is encouraged to improve the diagnosis and management of STH infection in animals. Further research is necessary to improve the LoD technique performance.

## Figures and Tables

**Figure 1 vetsci-11-00174-f001:**
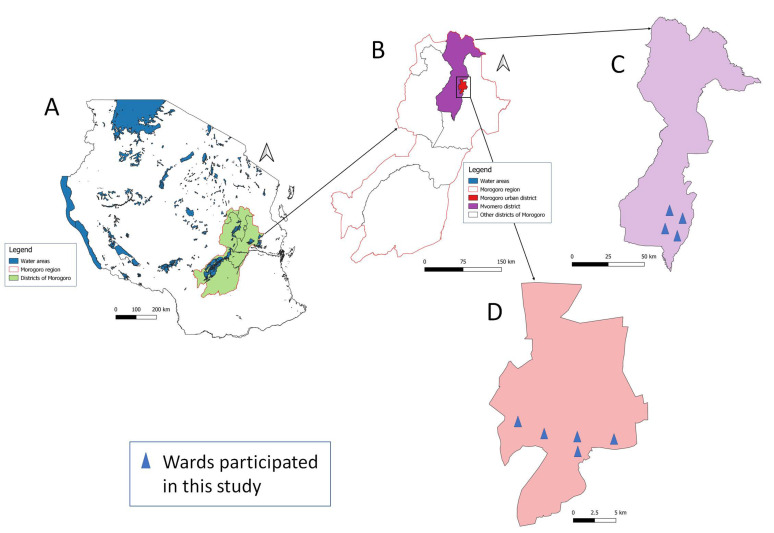
Map of Tanzania (**A**) showing the location of Morogoro region (**B**), Mvomero District (**C**), and Morogoro municipality (**D**).

**Figure 2 vetsci-11-00174-f002:**
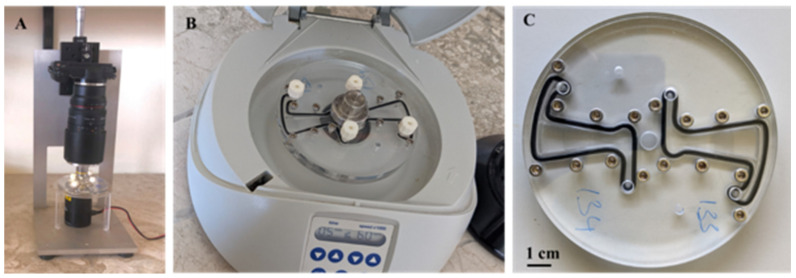
SIMPAQ setup. (**A**) SIMPAQ microscope with the disc ready for examination; (**B**) centrifuge; and (**C**) the lab-on-a-disc (LoD).

**Figure 3 vetsci-11-00174-f003:**
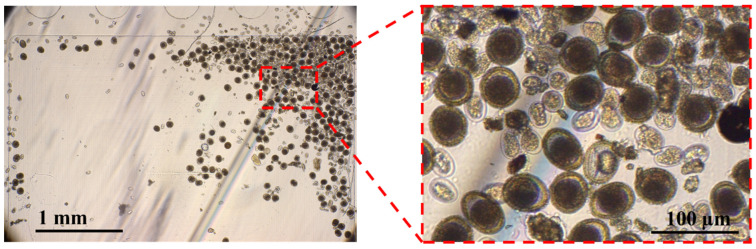
An image taken by the LoD technique, depicting STH species.

**Table 1 vetsci-11-00174-t001:** Overall LoD technique performance for both animal species.

Technique	Sensitivity	Specificity	PPV	NPV	TA
	% (95% CI)	% (95% CI)	% (95% CI)	% (95% CI)	Kappa Values
McM	93.51 (91.39–95.63)	60.89 (56.69–65.10)	81.91 (78.60–85.23)	83.21 (79.99–86.43)	0.5808
Flotation	93.18 (91.00–95.35)	59.67 (55.44–63.89)	81.14 (77.77–84.51)	82.44 (79.17–85.72)	0.5645

McM—McMaster; PPV—Positive predictive value; NPV—negative predictive value; TA—Test agreement; CI—Confidence interval.

**Table 2 vetsci-11-00174-t002:** Diagnostic performance of the LoD technique in domestic pigs.

Technique	Sensitivity	Specificity	PPV	NPV	TA
	% (95% CI)	% (95% CI)	% (95% CI)	% (95% CI)	Kappa Value
McMaster					
LoD	90.40 (86.7–94.2)	33.33 (27.3–39.3)	87.32 (83.1–91.6)	40.63 (34.4–46.9)	0.2558
Flotation	98.99 (97.7–100)	84.62 (80.0–89.2)	97.03 (94.9–99.2)	94.29 (91.3–97.2)	0.8720
Flotation					
LoD	90.10 (86.3–93.9)	34.29 (28.2–40.3)	88.78 (84.8–92.8)	37.50 (31.3–43.7)	0.2528
McM	97.0 (94.9–99.19)	94.3 (91.3–97.2)	98.99 (97.7–100.3)	84.62 (80.0–89.2)	0.8720

McM—McMaster; PPV—positive predictive value; NPV—negative predictive value; TA—test agreement; CI—confidence interval.

**Table 3 vetsci-11-00174-t003:** Diagnostic performance of LoD in dogs.

Technique	Sensitivity	Specificity	PPV	NPV	TA/Kappa Value
McMaster					
LoD	97.87 (96.2–99.6)	68.57 (63.1–74.0)	75.82 (70.8–80.8)	96.97 (95.0–99.0)	0.6651
Flotation	94.33 (91.6–97.0)	98.57 (97.2–100)	98.5 (97.1–99.9)	94.5 (91.9–97.2)	0.9288
Flotation					
LoD	97.78 (96.1–99.50)	65.8 (60.2–71.3)	72.5 (67.3–77.8)	96.97 (95.0–99.0)	0.6271
McM	98.52 (97.1–99.9)	94.52 (91.9–97.2)	94.33 (91.6–97.0)	98.57 (97.1–100)	0.9288

McM—McMaster; PPV—positive predictive value; NPV—negative predictive value; TA—test agreement; CI—confidence interval.

**Table 4 vetsci-11-00174-t004:** LoD performance for individual STH species in dogs (*n* = 281) and domestic pigs (*n* = 237).

STH Species	Strongyle		Ascarid		Whipworm	
Animal Species	Dog	Pig	Dog	Pig	Dog	Pig
Sensitivity	97%	88.36%	97.14%	71.43%	42.9%	72.73%
Specificity	69.59%	33.33%	93.90%	95.07%	96.4%	93.63%
PPV	74.14%	83.92%	69.39%	47.62%	23.08%	64.86%
NPV	96.26%	42.11%	99.57%	98.15%	98.51%	95.50%

PPV—positive predictive value; NPV—negative predictive value.

**Table 5 vetsci-11-00174-t005:** Overall LoD technique performance for individual STH species (*n* = 518).

STH Species	Strongyle	Ascarid	Whipworm
Sensitivity	91.93%	89.80%	67.50%
Specificity	60.71%	94.46%	95.19%
PPV	79.36%	62.86%	54.00%
NPV	82.07%	98.88%	97.22%

PPV—positive predictive value; NPV—negative predictive value.

## Data Availability

The data presented in this study are available on request from the corresponding author.

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
