# Peer review of "Evaluation of Lab-on-a-Disc Technique Performance for Soil-Transmitted Helminth Diagnosis in Animals in Tanzania"

_vetsci, 2024, doi:10.3390/vetsci11040174_

Round 1

Reviewer 1 Report

Comments and Suggestions for Authors

The authors compared two classical coproscopical (flotation and McMaster technique) methods with the lab-on-a-disk (LoD) technik and used 237 pig and 281 dog faecal samples collected in two districts of the Morogoro region of Tanzania. Soil transmitted helminth (STH) eggs (ascarids, ancylostomids, Trichurids and Capillariids) were chosen as indicator parasites. Hyostrongylus rubidus and Strongyloides ransomi would be two more STH species that might be present in pigs under tropical conditions.

The percentage of positive sample from a total of 518 (dog and pig) combined for flotation, McMaster and LoD techniques was 65.04, 65.44 and 74.0% respectively. Sensitivity and specificity as well as positive predicted value and negative predicted values as well as 95% confidence intervals were calculated to compare the three methods.

The main question is, are these three methods comparable, since a different quantity of faeces was used: for the flotation method and for the McMaster method 3g, for the Lod 1g.  What kind of McMaster chamber was used? There are a number of modifications on the market. The McMaster method was performed in an incorrect way since 42 ml of flotation fluid was used. The amount of liquid taken for filling the chamber has to be taken from an equal total amount of the filtered faecal-flotation solution. Thus, after filtration, the vessel from which the chamber is filled has to be filled to an equal level. In order to have the factor 100, the vessel has to be filled to a level of 60 ml. This would be the case if the chamber has two fields with a capacity of 0.1 ml per field.

The flotation method in other parts of the world is done differently by pouring the mixture of faeces and flotation solution through a sieve into a test tube. The tube is covered with a coverslip that has contact to the liquid column. After 20 min of flotation the coverslip is removed and placed on a slide and the slide is evaluated under the microscope.

The LoD method is complicated, and the amount of work is too high for a routine veterinary diagnostic laboratory.

It is not mentioned what kind of flotation solution with which density is used.

Most of the cited journals in the reference list are abbreviated, others not.

Reviewer 2 Report

Comments and Suggestions for Authors

General comment: Investigation of better diagnostic method is required for STHs. The study was done under good background and with enough number of materials. There are two critical issues for this kind of research on diagnosis evaluation.

The first is how to set the gold standard. The present study evaluated diagnostic efficacy of LoD compared with findings of McMaster test and simple flotation method. Both of them are known to be limited for diagnosis of STHs especially in low intensity infections.

The second one is proportion of samples with low burden of infection. Any diagnostic method may show good efficacy when subjected samples are with high EPGs.

Therefore, it is necessary to analyze data after resetting of gold standard and after grouping of samples by EPGs.

Comments on the Quality of English Language

A few sentences of language editing are required.

L 68: methods

L178, 179: floatation vs. flotation

L190-191: Check grammar 

Author Response

I have attached the the response to reviewer's comment in a word document 

Regards 

Reviewer 3 Report

Comments and Suggestions for Authors

The work presented is essentially sound, although it would require significant editing to bring it up to a publishable standard.  

My main issue with the paper is that the technique has already been published - and validated - by the Belgian group responsible for it's development.  

Unfortunately, therefore, I do not see this work as adding a significant advancement to the field.

Comments on the Quality of English Language

The standard of English is not appropriate for publication.

Round 2

Reviewer 1 Report

Comments and Suggestions for Authors

Just two comments regarding the references for two methods.

For the flotation method, three references were given (18,19,20). Reference 18 described the simple flotation while in reference 20, the flotation in test tubes was done. Better not to cite reference 20.

For the McMaster method references 18,21,22 were cited. While reference 18 used saturated NaCl solution, reference 22 used Sheater solution with a much higher density of 1.27. Better not to cite reference 22,

Author Response

We thank the reviewer for pointing this out.

Reference 20 and 22 were removed from the text and the references list as suggested

Regards 

Reviewer 3 Report

Comments and Suggestions for Authors

I must admit that I still have a few issues with the similarity of this manuscript to previous work (including some of the same authors).  However, now that they have changed the focus away from including children, to only dogs and pigs, I'm comfortable with it being published.  

Comments on the Quality of English Language

The standard of English does need to be addressed, as it is not sufficiently good for publication at the moment.

Author Response

We thank the reviewer for the comments.

The entire manuscript has been reviewed and corrected by microFlow researcher Matthieu Briet who is a native English speaker. 

Regards